# Resource

# MICA: a multi-omics method to predict gene regulatory networks in early human embryos

Gregorio Alanis-Lobato[1],* , Thomas E Bartlett[2],* , Qiulin Huang[1,3],* , Claire S Simon[1], Afshan McCarthy[1], Kay Elder[4] , Phil Snell[4], Leila Christie[4], Kathy K Niakan[1,3,5,6]

**Recent advances in single-cell omics have transformed characterisation of cell types in challenging-to-study biological contexts. In contexts with limited single-cell samples, such as the early human embryo inference of transcription factor-gene regulatory network (GRN) interactions is especially difficult. Here, we assessed application of different linear or non-linear GRN predictions to single-cell simulated and human embryo transcriptome datasets. We also compared how expression normalisation impacts on GRN predictions, finding that transcripts per million reads outperformed alternative methods. GRN inferences were more reproducible using a non-linear method based on mutual information (MI) applied to single-cell transcriptome datasets refined with chromatin accessibility (CA) (called MICA), compared with alternative network prediction methods tested. MICA captures complex non-monotonic dependencies and feedback loops. Using MICA, we generated the first GRN inferences in early human development. MICA predicted co-localisation of the AP-1 transcription factor subunit proto-oncogene JUND and the TFAP2C transcription factor AP-2γ in early human embryos. Overall, our comparative analysis of GRN prediction methods defines a pipeline that can be applied to single-cell multi-omics datasets in especially challenging contexts to infer interactions between transcription factor expression and target gene regulation.**

# Introduction

After the fusion of the oocyte and sperm, the zygote undergoes a series of cell divisions until it forms a blastocyst before implantation into the uterus. A human blastocyst is formed of a fluid-filled cavity and ~200 cells that comprise three distinct cell types: the trophectoderm (TE), which gives rise to fetal components of the

placenta; the primitive endoderm (PE), which forms the yolk sac; and the pluripotent epiblast (EPI), which gives rise to the embryo proper (Blakeley et al, 2015). The specification of these three lineages represents the earliest cell fate decisions in humans. Understanding the molecular mechanisms that regulate these decisions is important for applications including stem cell biology, regenerative medicine, and reproductive technologies (Niakan et al, 2012).

We do not yet understand how cell fate specification is regulated in the human embryo. Transcription factor (TF) and target gene regulatory interactions associated with cell fate specification in this context would be informative and has not been elucidated. Defining the gene regulatory networks (GRNs) associated with a given cell type at a distinct time in development facilitates characterisation of cell type identity and prediction of the transcriptional regulators and cis-regulatory DNA sequences that may underlie cell fate specification (Davidson & Erwin, 2006; Materna & Davidson, 2007; Peter & Davidson, 2011). Challenges to determining GRNs in early embryos include the small number of cells that contribute to the distinct cell lineages of the human embryo at these early stages of development. Moreover, although single-cell omics technologies facilitate the characterisation of genome-wide gene expression and chromatin accessibility changes in human blastocysts (Yan et al, 2013; Blakeley et al, 2015; Petropoulos et al, 2016; Li et al, 2018; Liu et al, 2019), it is unclear if resolution at this level would allow for accurate predictions of GRNs contributing to cell type specific identity.

The most common computational approach to infer TF-gene regulatory interactions is to model the expression of each target gene as either a linear or non-linear combination of the expression of a set of potential regulators (e.g., the TFs in the dataset of interest) (Dobra et al, 2004; Marbach et al, 2012). The "target gene" method uses advanced regression methods, such as sparse penalised regression or random forests, to model the expression level of a particular "target" gene (the response), conditional on the expression levels of a set of other genes (predictors, such as TFs)

[1]Human Embryo and Stem Cell Laboratory, The Francis Crick Institute, London, UK   [2]Department of Statistical Science, University College, London, UK   [3]Department of Physiology, Development and Neuroscience, The Centre for Trophoblast Research, University of Cambridge, Cambridge, UK   [4]Bourn Hall Clinic, Cambridge, UK   [5]Wellcome – Medical Research Council Cambridge Stem Cell Institute, Jeffrey Cheah Biomedical Centre, University of Cambridge, Cambridge, UK   [6]Epigenetics Programme, Babraham Institute, Cambridge, UK

Correspondence: kkn21@cam.ac.uk; g.alanis.lobato@gmail.com; thomas.bartlett.10@ucl.ac.uk
*Gregorio Alanis-Lobato, Thomas E Bartlett, and Qiulin Huang contributed equally to this work

(Huynh-Thu et al, 2010; Haury et al, 2012). By combining these local network model fits genome-wide, with each gene taking a turn as the response or "target gene" (Dobra et al, 2004), the GRN is constructed. The linear regression model has previously been used in combination with chromatin accessibility to infer GRNs in human brain organoids (Fleck et al, 2023). Bayesian networks also model the expression of each gene conditional on a set of regulating genes (i.e., parents in the network) by computing how likely it is that each TF in the dataset was responsible for the expression of a certain gene (Tipping, 2001; Pe'er, 2005; Marbach et al, 2012; Kamimoto et al, 2023). The Bayesian/Bagging Ridge model in combination with chromatin accessibility has been used effectively to predict the consequence of gene perturbations and to under-stand cell fate transitions in the context of development and cellular reprogramming (Argelaguet et al, 2022 *Preprint*; Kamimoto et al, 2023). Of note, when fitting Bayesian models, it is usually necessary to assume global compatibility (of the local network models), which in turn typically requires that the graph contains no loops (acyclicity). This is an assumption which is challenging for biological systems, where feedback loops are a persistent feature (Brandman & Meyer, 2008). In another important group of methods, TF-gene (or more generally gene–gene) interactions are ranked based on a pairwise similarity measure of their expression across cell samples, such as a variant of correlation or mutual information (Daub et al, 2004; Krishnaswamy et al, 2014). Again, alternatives are available that consider conditional statistics (e.g., partial correla-tion or partial information decomposition) (Chan et al, 2017). Finally, a range of "black-box" machine learning methods is available that models the expression of each gene in hard to visualise ways using for example sophisticated non-linear combinations of TF expres-sion values (Wang et al, 2020; Shu et al, 2021).

Although GRN inference methods have been comprehensively compared and developed for whole-tissue or bulk transcriptome analysis (Marbach et al, 2012; Cahan et al, 2014; Chai et al, 2014; Morris et al, 2014; Thompson et al, 2015), the interest in cell type–specific regulatory networks has prompted the application of these methods to single-cell RNA-seq (scRNA-seq) datasets (Aibar et al, 2017; Chen and Mar 2018; Fiers et al, 2018; Mochida et al, 2018; Iacono et al, 2019; Pratapa et al, 2020; Kang et al, 2021; Nguyen et al, 2021; Stone et al, 2021 *Preprint*; Badia-i-Mompel et al, 2023). These previous studies showed that the approaches with the best overall performance belong to the regression and mutual information categories and highlighted the challenges of performing GRN inference on single-cell datasets. This is attributed mainly to the heterogeneous and stochastic nature of gene expression in individual cells (Nguyen et al, 2021). Of note is that even state-of-the-art methods that have been evaluated on well-defined benchmark datasets predict a considerable number of false-positive interactions (Marbach et al, 2012; Pratapa et al, 2020; Kang et al, 2021; Nguyen et al, 2021; Stone et al, 2021 *Preprint*). Most of these false predictions originate from indirect associations, for example a path $a \rightarrow b \rightarrow c$ can result in the prediction of $a \rightarrow c$ even if there is no direct link between those nodes (i.e., a "transitive edge" in the network). Several methods to eliminate such effects have been proposed (Margolin et al, 2006; Barzel & Barabási, 2013; Feizi et al, 2013; Chan et al, 2017; Wang et al, 2018) but they can be computa-tionally demanding and sensitive to the choice of hyperparameters (Feizi et al, 2013).

Given the stage of development of human preimplantation embryos and their precious nature, together with the restrictions on such research in some countries, the omics datasets from human blastocysts are very small compared with those from other biological contexts. This makes it challenging to mine these datasets using GRN inference methods, which require a sufficiently large number of cells to produce reproducible results (Pratapa et al, 2020; Kang et al, 2021). This tight restriction on sample sizes places corresponding restrictions on statistical power and means that the optimal statistical network inference methodology may be specific to this context. Here, we assessed whether the integration of other types of omics datasets with transcriptomic-based predictions could help reduce indirect TF-gene relationships and thereby produce more reliable GRNs in this setting, especially with re-stricted sample sizes. We used a low number of cell samples in high-quality omics data from early human embryos at the blas-tocyst stage (6–7 d post-fertilisation) (Yan et al, 2013; Blakeley et al, 2015; Petropoulos et al, 2016) to evaluate the plausibility of our predictions: this has the advantage of being a biological context with only three well-defined cell types to evaluate the plausibility of our predictions.

The aims of this work were (i) to evaluate whether it is possible to infer reliable cell type–specific regulatory networks for each one of the cell types in the early human embryo in spite of the size of the available omics data, (ii) to determine if the integration of chromatin accessibility data with transcriptome analysis would better inform predictions of GRNs, and (iii) to predict and val-idate previously unidentified gene regulations in the human blastocyst. Overall, we demonstrate that the available single-cell transcriptomic data were most robustly analysed by a non-linear mutual information-based inference method which had been refined with chromatin accessibility data (MICA). The resulting analysis predicted the first GRN in human preimplantation de-velopment and showed that the interactions were consistent with the transcripts and proteins that are known to be enriched in specific lineages. MICA predicted a novel putative regulatory interaction between the TFAP2C transcription factor AP-2γ and the AP-1 transcription factor subunit proto-oncogene JUND in human preimplantation embryos. Overall, we propose that MICA will be an informative method to make GRN predictions in other challenging-to-study biological contexts with limitations in the number of cells that can be analysed.

## Results

### Assessing the conditions for optimal GRN inference on synthetic data

We generated synthetic gene expression data with known ground-truth GRN structure for a variety of sample sizes (see the Materials and Methods section) to determine the characteristics of the transcriptomic data on which GRN inference methods perform best and, importantly, to compare these conditions with our human blastocyst scRNA-seq datasets (see Fig 1A). We then applied four different GRN prediction strategies to the simulated transcriptomes

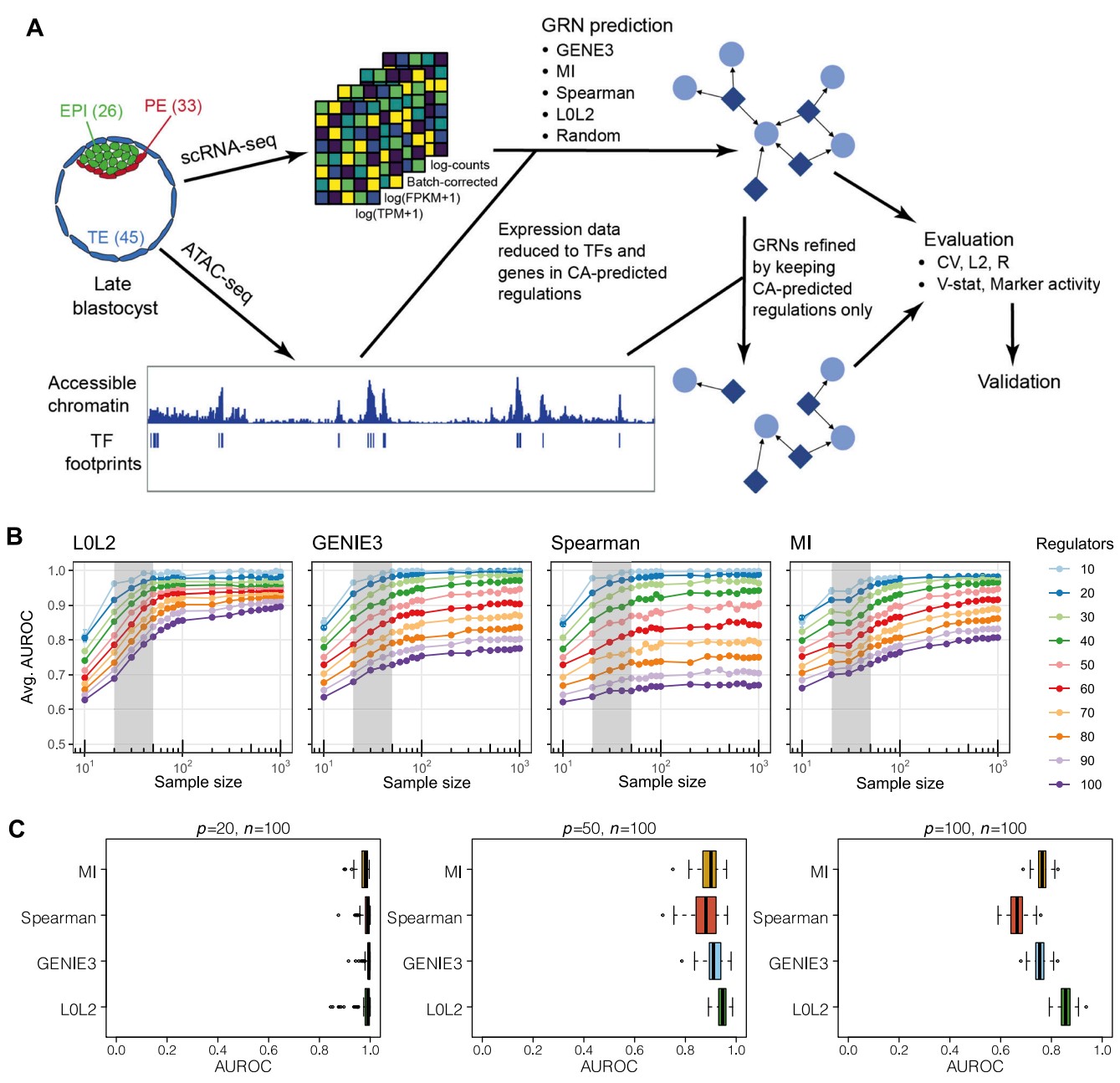

**Figure 1.  An overview of the regulatory network inference, evaluation, and validation framework and inference on simulated single-cell RNA-seq (scRNA-seq) data.**
**(A)** Footprinting analysis was applied to ATAC-seq data from human embryos at the late blastocyst stage to identify potential transcription factor (TF) binding sites and TF-gene regulations. The TFs and genes in this list of chromatin accessibility (CA)-predicted regulations were used to refine the size of scRNA-seq data from each of the cell types in embryos where cells were collected at the same developmental stage (EPI, Epiblast, 26 cells; PE, Primitive Endoderm, 33 cells; TE, Trophectoderm, 45 cells). Four gene regulatory network (GRN) inference approaches and a random approach were applied to the scRNA-seq data. The scRNA-seq data were normalised using four different methods for comparison (log(TPM + 1), log(FPKM + 1), batch-corrected, and log-counts).The reproducibility and biological context of the predicted GRNs were evaluated using several statistical tests. GRNs refined by keeping CA-predicted regulations only were also evaluated. Squares represent genes that regulate other genes, and circles represent genes that are regulated by other genes. **(B)** Comparison of different GRN inference methods (L0L2, GENIE3, Spearman correlation, and MI) to recover the ground-truth GRN structure from simulated gene expression data as measured with the average area under the receiver operating characteristic curve (AUROC) from 100 simulations. The average AUROC is shown as a function of different sample sizes (*n* = 10–1,000) and the number of potential regulators of each gene in the simulated datasets is also varied from 10 to 100. The range of sizes of the three human blastocyst datasets that we analysed is highlighted in grey as a reference. Error bars correspond to standard errors of the mean. **(C)** Box and whisker plots show comparison of AUROC values for all simulations for each method with number of samples *n* = 100 and number of regulators *P* = 20, 50, and 100.

(see the Materials and Methods section for more details). (1) We applied GENIE3 (Huynh-Thu et al, 2010; Aibar et al, 2017), a random forests regression approach which takes into account expression levels of the regulated or target gene and is used for regulatory network inference in the SCENIC GRN prediction pipeline (Marbach et al, 2012; Aibar et al, 2017; Pratapa et al, 2020; Kang et al, 2021; Nguyen et al, 2021; Stone et al, 2021 *Preprint*). (2) We compared this to the target gene approach using the L0L2 sparse penalised regression model (Tibshirani, 1996; Hazimeh & Mazumder, 2018 *Preprint*) which minimizes the total size of the linear model coefficients so that a minimal set of regulatory TFs receive the largest linear model coefficients. (3) We also compared with the correlation coefficient as a reference: we used Spearman's rank correlation coefficient (Fieller et al, 1957) as we expect non-linear associations in the data. (4) Lastly, we applied a non-linear alternative measure of pairwise association based on mutual information (MI) that we speculated may be more appropriate for non-linear response functions in single-cell data (Faith et al, 2007; Krishnaswamy et al, 2014) (Fig S1A). For the MI method, we used the empirical distribution of the MI values for each gene (Faith et al, 2007). We also varied the number of transcriptional regulators of each target gene in the synthetic data to study the impact of this parameter on their performance. Further details of the analysis methods and step-by-step code are also available at https://github.com/galanisl/early_hs_embryo_GRNs.

Two main conclusions emerged from these simulations. First, using synthetic datasets, the performance of all methods increases from the sample size of $n$ = 10–1,000, as assessed statistically by the area under the receiver operating characteristic curve (Fig 1B). However, we noted that the prediction accuracy of the GRNs only marginally improves as the number of cell samples surpasses $n$ = 100 (Fig 1B). Therefore, although sample size is important, increasing this beyond a threshold sample size does not further improve inference of the predicted transcriptional regulations. We furthermore note that low sequencing depth (such as in 10x Genomics single-cell studies) may increase this optimal value of $n$. Second, limiting the number of potential transcriptional regulators of a target gene positively impacts the ability of a chosen inference method to recover the ground-truth GRN (Fig 1B and C). From our simulations, 50 or fewer TFs yield the highest area under the receiver operating characteristic curve values for $n$ < 100 (Fig 1B). This would be a reasonable prediction for a given gene in a specific cell type based on our analysis of low-input chromatin accessibility (CA) data from the human blastocyst (Liu et al, 2019), where the median number of TF motifs per gene is 35 for the inner cell mass (ICM) and 40 for the TE (Fig S1B).

Based on these results and given the size of our human blastocyst datasets (Fig 1C EPI: 26 cells, PE: 33 cells, TE: 45 cells with 1,366 TFs among 25,098 genes [Yan et al, 2013; Blakeley et al, 2015; Petropoulos et al, 2016]), we reasoned that TF-gene interactions predicted from experimentally collected gene expression data could be refined with complementary, context-specific epigenomic datasets. Specifically, we used low-input CA analysis of human blastocyst TE cells and the ICM, comprised of EPI and PE cells, to refine the GRNs of the respective cell types with putative cis-regulatory interactions (Liu et al, 2019). Peak calling and annotation were performed with nf-core/atacseq to identify regions of

open chromatin (Ewels et al, 2020). TF motif enrichment analysis in these open regions of open chromatin was identified using rgt-hint with TF binding models from HOCOMOCO and JASPAR (Li et al, 2019). Finally, the open chromatin regions enriched for TF motifs along with the predicted downstream target genes were determined based on the nearest transcriptional start sites (most distances ranging from −5 to 10 kbp). This narrowed down the number of genes (to 12,780 for EPI and PE, 12,981 for TE) and TFs (514) in the datasets, thus bringing our sample sizes and potential TFs per gene to the ranges suggested by our simulations.

## Statistical evaluation of inferred human blastocyst GRNs

We next integrated the single-cell transcriptome data (Fig S1C) from human blastocysts generated using the SMART-seq2 library preparation protocol (Yan et al, 2013; Picelli et al, 2014; Blakeley et al, 2015; Petropoulos et al, 2016) where fewer cells are sequenced but better transcript coverage and sequencing depth is obtained (~15,000 genes and ~7 million reads per cell) compared with more conventional scRNA-seq methods, such as 10x Genomics (~3,000 genes and ~50,000 reads per cell) (Wang et al, 2021). Because of the lack of genome-scale experimentally derived GRNs in the human embryo context, we evaluated the robustness of the predictions made by four inference approaches (L0L2 regression, GENIE3, Spearman correlation, and MI) using three different strategies (see the Materials and Methods section). Because we did not find reports about the impact of gene expression normalisation choice on GRN predictions, we also assessed this parameter and considered the application of GRN predictors to log(TPM + 1), log(FPKM + 1), log-count, and batch-corrected expression data.

We calculated a reproducibility score $R$ for each putative regulatory interaction after the application of the network inference methods to human blastocyst data (see the Materials and Methods section) and investigated the distribution of the reproducibility values for the top 100,000 predicted edges (Figs 2A and S2). The reproducibility estimator $R$ estimates the posterior probability of seeing a network edge given the data; it quantifies the robustness or stability of the inference of this network edge. The accuracy quantified by the $R$ reproducibility statistic relates to the stability of the model predictions to perturbation of the data. Fig 2A reports the difference between the median of this distribution and the median of the distribution produced by a predictor that generates a random ranking of all possible TF-gene interactions ($\Delta R$). We found that the most reproducible regulatory interactions were inferred by MI followed by a filtering process in which only TF-gene associations supported by CA data were considered in the final network (MICA) (Fig 2A). As predicted by our simulations, the size of each cell type–specific dataset had a clear impact on GRN inference with methods such as GENIE3 producing more reproducible interactions in the TE. Interestingly, most inference methods produced better results with gene expression values following log(TPM + 1) or log(FPKM + 1) normalisation (Fig 2A).

We also assessed robustness at the level of network structures, features, or subnetworks. To do so, we randomly split each dataset (EPI, PE, and TE) into two groups with the same number of cells, then applied the inference methods to each one (with or without CA refinement), and finally benchmarked them either focussing on the

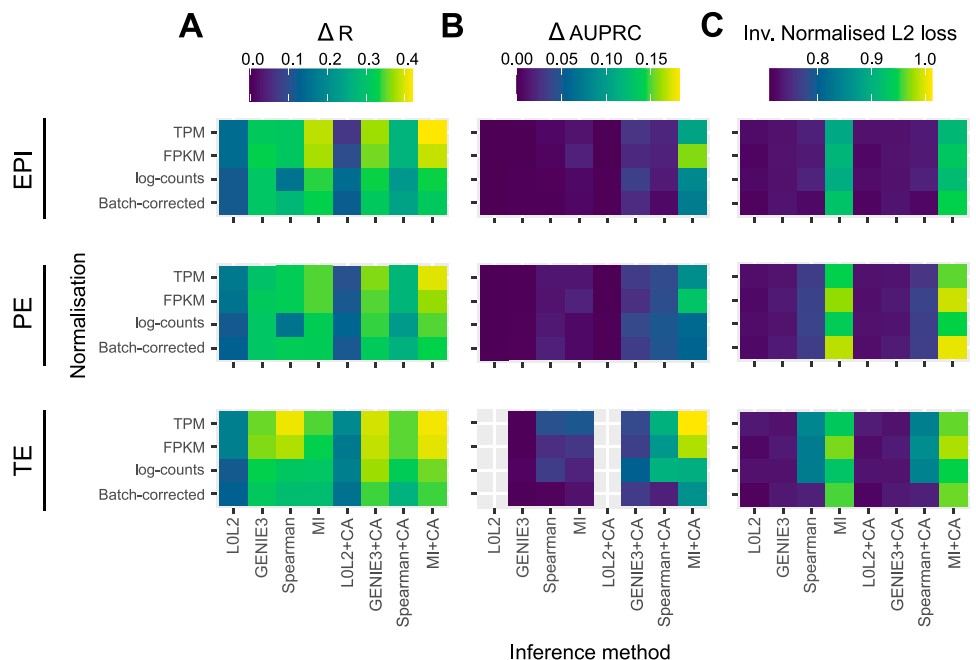

**Figure 2. Statistical comparison of four different methods to infer early human embryo gene regulatory networks (GRNs).** Robustness of the GRNs predicted by four inference methods with or without chromatin accessibility (CA) refinement was evaluated by three different metrics. **(A)** ΔR measures the difference between the median edge-level reproducibility for a GRN inference method and a random predictor. **(B)** ΔAUPRC quantifies the extent to which the interactions inferred from half of a dataset coincide with the top 1% interactions inferred from the second half. This twofold cross-validation experiment was repeated 10 times and compared with a random predictor. **(C)** The inverse of the normalised L2 loss between the network scores from the twofolds was computed. L0L2 did not converge when applied to the TE dataset. All data used was generated using the SMART-seq2 sequencing technique.

top regulatory interactions (early recognition) or on all regulatory interactions (see the Materials and Methods section). We repeated this 10 times for 10 random splits of the data. For the first comparison, we identified the top 1% (by network score) of all possible regulatory interactions from each group, and then quantified the overlap of these top interactions between the groups using the area under the precision-recall curve (AUPRC) (Figs 2B and S3). By comparing the differences between the median of 10 resulting AUPRCs for each method to the random predictions described above, we found that MICA outperformed the other network prediction approaches and gave the highest difference in AUPRC when applied to log(TPM + 1)- and log(FPKM + 1)-normalised expression values (Fig 2B). For the second comparison, we divided the data into two portions and then calculated a normalised L2 loss between the network scores over the whole network, rather than just the top 1% of interactions (Figs 2C and S4). We found that the inverse value of this metric also confirms the robustness of MICA (Figs 2C and S4).

Overall, our statistical evaluation strategies showed that the most robust GRN inference method to analyse the limited number of human blastocyst cells was MICA. Importantly, if transcriptome-based predictions are not refined with CA data, most methods perform just slightly better than random (Figs 2, S2, S3, and S4), underscoring the importance of integrating multi-omics analysis in inference models. It was also important to assess the impact of gene expression normalisation on GRN prediction because we observed apparent effects in our benchmarks depending on the normalisation method used, with log(TPM + 1) or log(FPKM + 1) being the most suitable gene expression units in this context.

## Association of inferred GRNs to human blastocyst cell lineages

We next evaluated the inferred GRNs to determine if they could recapitulate interactions of molecular markers of the three cell

types that comprise the human blastocyst. First, we computed the overlap between the GRN edges predicted by each inference method for each blastocyst cell type to identify interactions associated with the EPI, PE, and TE, as well as the interactions common to the three cell types (see the Materials and Methods section). We then used the out-degree of NANOG, GATA4, and CDX2, TFs which respectively mark the EPI, PE, and TE (Fig 3A and B), as a proxy for their activity in each one of the four networks. Our prediction is that these TFs should be actively regulating genes in the cell type–specific networks and participate in only a few interactions in the common GRN based on their known expression pattern in the blastocyst and function in other mammalian contexts such as the mouse (Arceci et al, 1993; Chambers et al, 2003; Mitsui et al, 2003; Strumpf et al, 2005; Dietrich & Hiiragi, 2007; Niakan & Eggan, 2013; Roode et al, 2012). The common network did not show marker activity in the predicted GRNs (Figs 3B, S5, S6, S7, and S8), which is expected given that the expression of these markers is known to be mutually exclusive at this stage. The expected pattern for NANOG (active in the EPI and inactive in the PE and TE, Fig 3A) was observed in the GRNs with TF-gene interactions supported by CA data and inferred with GENIE3, Spearman correlation, and MI when applied to batch-corrected, FPKM, and TPM data with some instances of the non-refined L0L2, GENIE3, and Spearman GRNs also matching the expected pattern (Fig 3B). In the PE, the expected activity pattern for GATA4 (active in the PE and inactive in EPI and TE, Fig 3A) and lack of detectable networks for NANOG and CDX2 were predicted by applying GENIE3+CA, Spearman correlation+CA, and MICA to the FPKM and TPM datasets (Fig 3B). Finally, the TE-expected pattern for CDX2 (active in the TE and inactive in the EPI and PE) and lack of detectable network for NANOG and GATA4 (Fig 3A) was only observed when using Spearman correlation or MICA on log-counts and batch-corrected data, MICA or MICA+CA on FPKM or TPM data (Fig 3B). L0L2, with or without CA refinement, consistently performed

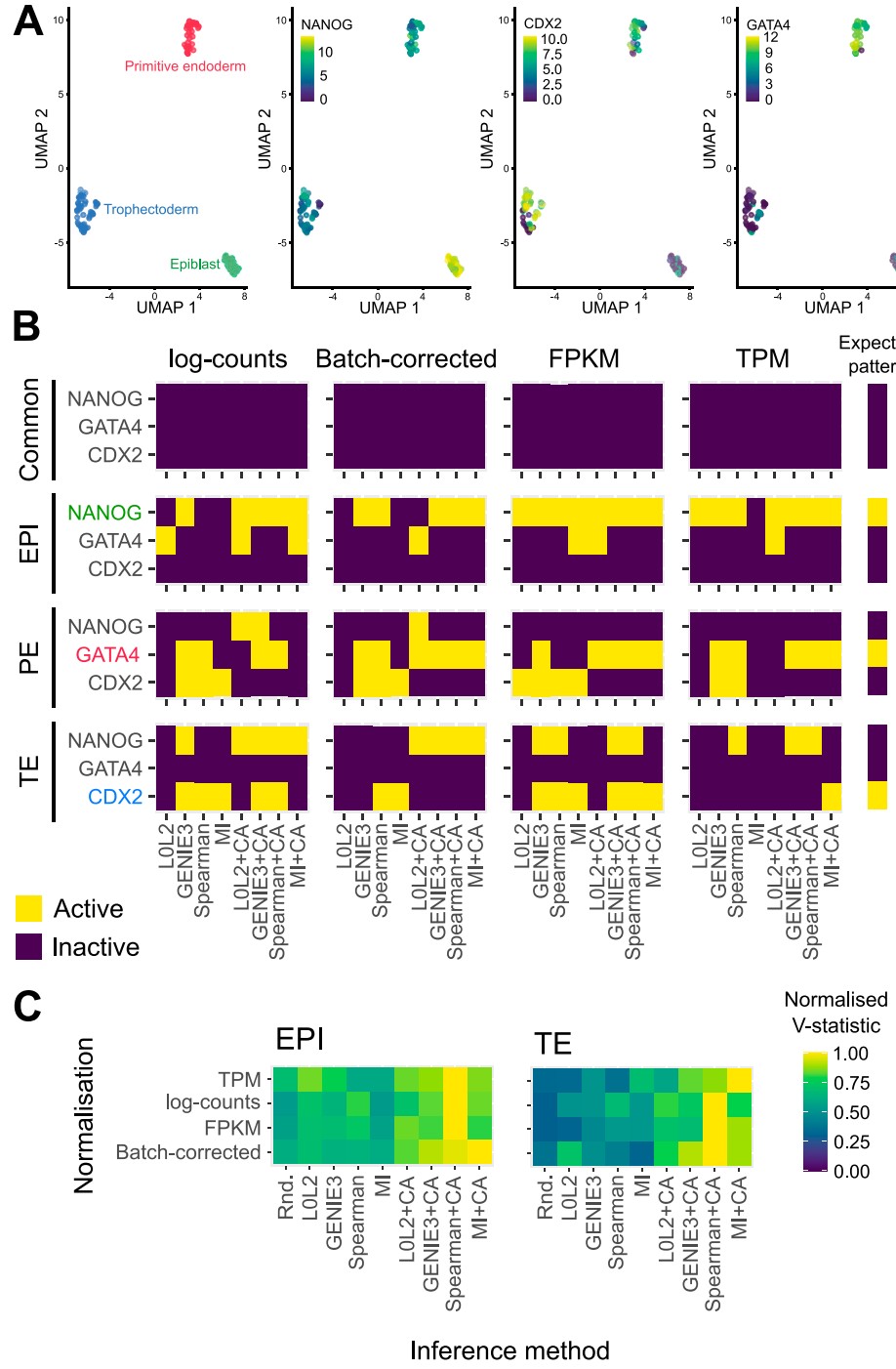

**Figure 3. Evaluation of the biological relevance of inferred gene regulatory networks (GRNs) in the context of early human development.**
**(A)** Two-dimensional UMAP representation of the single-cell RNA-seq data from each of the three cell types in the early human at the blastocyst stage. The expression of marker genes for each cell type is also represented in UMAP space: NANOG is used as an epiblast marker, CDX2 as a trophectoderm marker, and GATA4 as a primitive endoderm marker. **(B)** Marker activity in the cell type–specific GRNs (EPI, epiblast; PE, primitive endoderm; TE, trophectoderm) and the network common to the three cell types in the human blastocyst. The expected pattern, that is, the marker that was expected to be active in each GRN is shown for reference. **(C)** Normalised V-statistic across datasets and inference methods. V corresponds to the number of gene sets that were enriched at a significance level of 10% in a gene set enrichment analysis performed with the genes involved in the top-500 regulations by the 25 most-connected TFs in the EPI or TE GRNs. All sequencing was performed using the SMART-seq2 technique.

poorly at predicting GRNs compared with the other methods used. Log-count normalisation failed to recover cell type–specific GRNs for these TFs.

We next manually curated a list of gene sets representing the most relevant pathways and biological processes in the EPI (e.g., regulation of pluripotency) and the TE (e.g., embryonic placenta development) to perform a gene set enrichment analysis with the genes involved in the 500 regulatory interactions with the best

prediction scores in the EPI and TE GRNs of the 25 most active TFs (Tables S1 and S2, see the Materials and Methods section). We restricted this analysis to the EPI and TE because similar lists of gene sets were not available for PE, where it is currently unclear which pathways are most relevant to this cell type. Next, we computed a validation score *V* as the number of relevant gene sets that were enriched at a significance level of 10% (*P* < 0.1) in the EPI and TE GRNs (Fig S9). We note that a low significance level was set

because the resulting inferences are aggregated, mitigating the effect of false positive hits for individual gene groups. To facilitate the comparison between *V* scores across GRNs from different inference and normalisation methods, we normalised this score to the maximum score attained at the cell type level (Fig 3C). We found that the CA-refined data agree better with the cell type–specific gene sets for both the top-predicted EPI and TE interactions, that is, the interactions with the highest prediction scores (Fig 3B). The GRNs inferred by Spearman+CA produced the highest *V* scores in the EPI and TE, and these were not impacted by the normalisation method used. The second-best inference method was MICA, with consistent *V* scores across normalisation methods. We note that both MICA and Spearman+CA are non-linear methods which do not involve regression, indicating that advanced regression methods may not be the most effective choice for biological discovery with these restricted sample sizes. Taken together, both biological evaluation metrics that we considered confirmed the importance of refining single-cell transcriptome–based GRN inferences with CA data and underscored the robustness of MICA in predicting GRNs from the analysed human blastocyst datasets. Based on these results, we decided to focus on the GRNs predicted by this method for the analyses presented in the following sections.

### Predicted TF networks for NANOG, GATA4, and CDX2

Using MICA, we constructed GRNs for TFs expressed in the EPI, PE, and TE. All MICA GRN predictions can be found on FigShare: doi.org/10.6084/m9.figshare.21968813. For visualization, the predicted GRN for each of the TFs is separated into target and regulator TF networks. Target networks contain a maximum of 25 top potential target TFs of the hub (or central) TF, whereas regulator networks include a maximum of 25 top TFs that potentially regulate the hub. The average expression of the network members across samples of the cell type of interest is represented by the size of the node. MI scores are represented by the thickness of the edges in the network and edge colour highlights. To further refine the MICA predictions, we used the Spearman's rank correlation coefficient between the expression levels of the source and target nodes across samples of the cell type of interest to define correlated or anti-correlated expression. Correlated or anti-correlated node pairs correspond to positive or negative Spearman's rank coefficient with *P*-value smaller than 0.1, whereas node pairs having *P*-value equals or larger than 0.1 were defined as uncorrelated.

Among the top NANOG targets, TFs RREB1, NCAO3, ZNF343, ZFP42, and NME2 are predicted to be positively regulated by NANOG (Fig 4A). ZFP42 is a pluripotency marker encoding the REX1 protein and has been shown to be a direct target of NANOG in mouse pluripotent stem cells (Shi et al, 2006). ZNF343 is a less well-characterized TF, but multiple NANOG ChIP-seq datasets in both naïve and primed human embryonic stem cells (ESCs) showed high binding score (MACS2 score > 1,000) in the proximal region of the transcription start site (TSS) of ZNF343 (Fig S10A; Barakat et al, 2018; Lyu et al, 2018; Chovanec et al, 2021), which suggests direct regulation of ZNF343 by NANOG. Interestingly, NME2 was previously predicted to be a regulator instead of a target of NANOG in mouse pluripotent stem cells using the TENET GRN inference method (Kim et al, 2021). This inconsistency may be due to the lack of chromatin accessibility data for robust

directionality inference in the TENET method, or a species difference. Indeed, some mouse naïve pluripotency regulators, such as ESRRB, which is also a direct target of NANOG in mouse pluripotent stem cells (Festuccia et al, 2012), were not expressed in the human EPI (Blakeley et al, 2015). Putative regulators of NANOG predicted from our MICA network analysis contain multiple KLF factors, including KLF3, KLF5, KLF9, and KLF16 (Fig 4B). Spearman correlation analysis suggest that KLF9 and KLF16 potentially down-regulate NANOG expression, based on their anti-correlated expression, whereas KLF3 and KLF5 are identified as potential regulators of NANOG by MICA, but not significantly correlated by Spearman correlation.

In the PE, GATA4 was predicted to positively regulate the expression of SP8, TET1, and SKIL whereas repressing the expression of ELF3, TFDP2 POGK, ZNF770, and NR2F2 (Fig 4C). It is also predicted to be a target of HNF1B and SALL4 and repressed by ETV4 and E2F6 (Fig 4D). These interactions have not been experimentally validated or inferred. However, NR2F2 was previously identified as a maturation marker of polar TE (Meistermann et al, 2021). The repression of NR2F2 by GATA4 predicted in the PE suggests a role for GATA4 in maintaining the PE cell identity by inhibiting the polar TE program. Furthermore, it has been shown that SALL4 is required for mouse PE-derived extra-embryonic endoderm cell derivation and knockout of SALL4 in these cells cause down-regulation of GATA4 (Lim et al, 2008). In addition, multiple endoderm genes such as GATA4, GATA6, and SOX17 were shown to be SALL4-bound genes by ChIP-seq (Lim et al, 2008). These findings are consistent with the inferred GATA4 regulatory network.

Interestingly, TBX3 is a target of CDX2 (Fig 4E), and the role of TBX3 has been implicated in trophoblast cell differentiation (Lv et al, 2019). CDX2 is predicted to be regulated by both KLF5 and KLF6, which are also molecular markers of the trophectoderm lineage (Fig 4F). KLF5 is necessary for trophectoderm formation in the mouse preimplantation embryo and is required for CDX2 expression in mouse ESCs (Ema et al, 2008; Lin et al, 2010). Interestingly, further analysis of the CDX2 interactions showed that TBX3 and FOXH1 are also potential positive regulators of KLF5 and KLF6. Positive feedback loops are predicted between CDX2, TBX3, and KLF5 as well as CDX2, FOXH1, and KLF6 (Fig S10B). Other key developmental regulators including HAND1, GATA3, and GATA6 are also predicted to regulate CDX2 expression (Fig 4F). This is consistent with the timing of GATA3 protein expression preceding that of CDX2 (Gerri et al, 2020). Further investigation will be needed to understand the differences between the CDX2-high and CDX2-low TE cells and how the positive feedback loops formed within the CDX2 network enhance and stabilize CDX2 expression in the CDX2-high population.

### Maintenance of TFAP2C expression by JUND in all lineages of human blastocyst

We next sought to determine if the MICA network modelling predicts novel interactions or associations between TFs in early human embryo development. We focused on TFAP2C as an example for the network comparison. TFAP2C is a molecular marker that is initially expressed in all the cells at the morula stage in mouse embryos and later specifically restricted to the TE at the blastocyst stage and it is not expressed in other lineages (Cao et al, 2015; Gerri et al, 2020). However, in human embryos, TFAP2C is expressed in the TE, EPI, and PE at the blastocyst stage (Blakeley et al, 2015) and has been shown

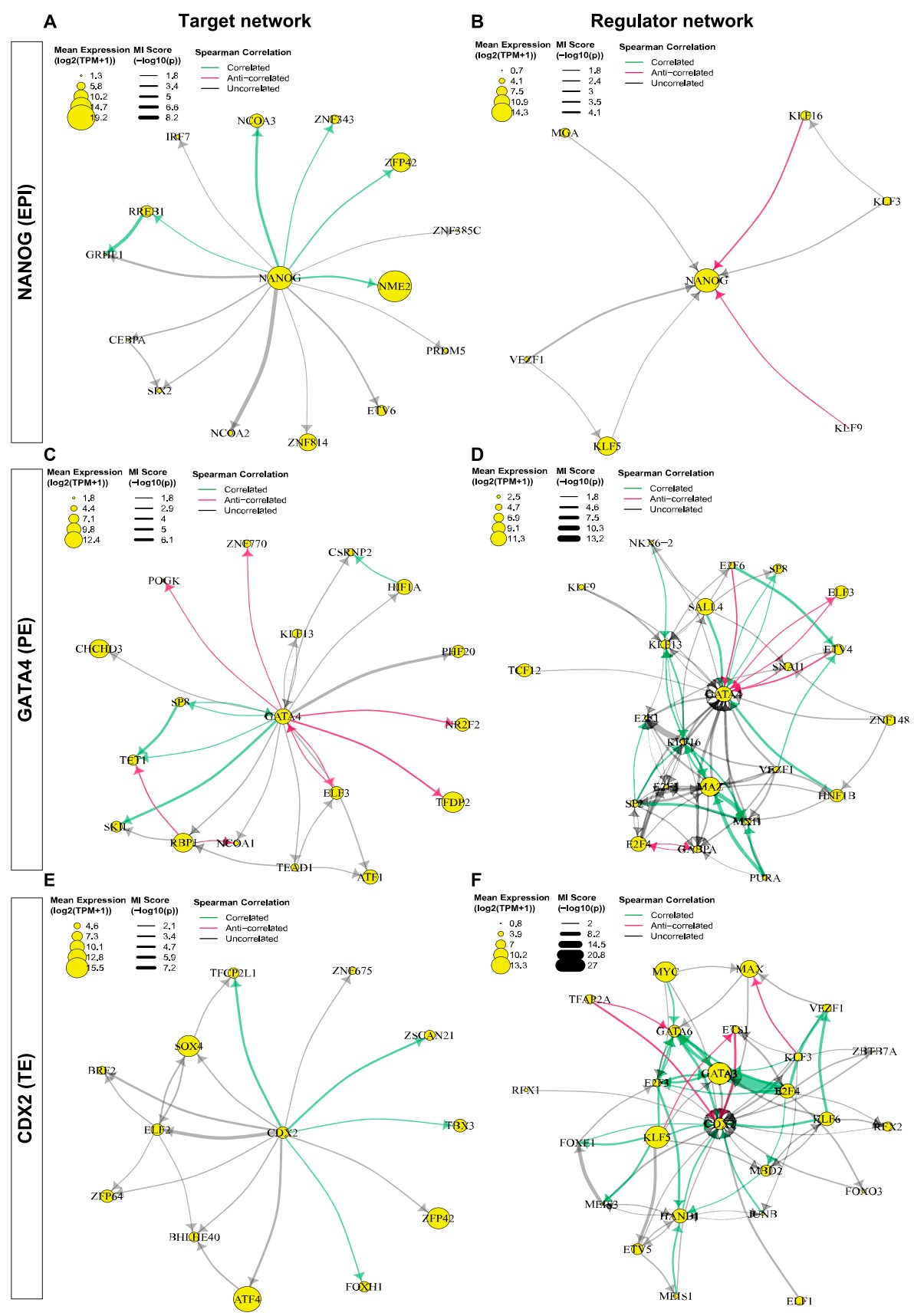

to maintain pluripotency in naïve human ESCs by regulating OCT4 expression (Chen et al, 2018; Pastor et al, 2018). By performing comparisons of TFAP2C networks in human EPI, TE, and PE cells, we found a conserved putative interaction consisting of TFAP2C, JUND, SOX4, and GCM1 (Fig 5A). In addition, all four factors showed significant positive correlations in their expression (Figs 5B and S11A and B). MICA predicts that JUND and SOX4 regulate TFAP2C, whereas TFAP2C targets GCM1 in all three lineages. In the EPI and PE, SOX4 and GCM1 formed a feedback loop. Interestingly, in the TE, the correlation between GCM1 and SOX4 is absent, and SOX4 is predicted to target JUND, which in turn may regulate TFAP2C. Our network predicts that interactions between these four TFs maintain TFAP2C expression in all three lineages.

To determine if TFAP2C and JUND protein expression is positively correlated in embryos, as predicted by MICA, we performed immunofluorescence analysis of human preimplantation blastocysts ~6.5 d after fertilisation. We observed that TFAP2C and JUND were co-expressed in cells of the human blastocysts (Figs 5C and S13A; n = 4). We next performed 3D nuclear segmentation and calculated the Pearson correlation coefficient between TFAP2C and JUND based on the DAPI-normalized protein intensity (Fig 5D). TFAP2C intensity showed a linear increase with the increase of JUND intensity in all analysed human embryos, and their intensities are highly correlated (0.68–0.85 Pearson correlation; $P < 0.001$; Fig 5D). Overall, this demonstrates the informativeness of the MICA network analysis to predict correlations and possible regulatory relationships between transcription factors that can be experimentally tested.

In addition to the MICA predictions, we also performed Spearman correlation analysis of the significant edges identified by the MICA analysis. Surprisingly, around half of the interactions are in non-linear fashion, highlighting the informativeness of MICA to capture complex non-monotonic dependencies. From the TFAP2C network predicted in the EPI lineage, we found that TFAP2C potentially regulates FOXO3 and ZFP42 in a non-linear manner (Fig S12). The expression of FOXO3 and ZFP42 seems to fit better with the expression of TFAP2C on an exponential curve rather than a linear line between TFAP2C and JUND (Fig S13B). This suggests that when analysing scRNA-seq or low-input multi-omics analyses similar non-linear correlations may exist and this may have biological significance. It would be interesting to know whether and which types of non-linear interaction predominate and the biological significance of these non-linear regulations.

## Discussion

The relationship between genome-wide transcriptomic and epigenomic changes and cell fate specification in human embryogenesis

is unclear. Studies of human preimplantation development rely on the donation of surplus embryos derived from assisted reproduction technologies, and the use of such embryos for research is tightly regulated and subject to significant limitations, such as a lack of ability to conduct such research in some jurisdictions (Niakan et al, 2012). In addition, the collection of single cells from such precious embryos is technically challenging and requires specialist expertise and micropipettes to disaggregate microscopic embryos. Therefore, currently available omics datasets from human blastocysts comprise only a few tens of samples per cell type and are therefore very limited. This contrasts with single-cell analyses in other cellular and developmental contexts that are based on tens of thousands of samples (or more) per cell type (Zheng et al, 2017).

Sample size is one of the most important considerations when selecting or designing statistical methodology, for example to infer networks of regulations of TFs and their target genes. Hence, GRN inference in omics data from human preimplantation embryos presents unique statistical challenges. In particular, methodology that can leverage information about gene regulations from small sample sizes is required for this context. On the other hand, the lack of heterogeneity in early human embryos compared with adult tissue makes this a good context in which to assess GRN inference methodologies because there is less unmeasured variability arising from environmental factors. To assess whether GRN inference method can be informative in this challenging-to-study context, we have systematically compared several popular methodologies. Furthermore, we have tested how incorporation of complementary cis-regulatory epigenomic data from ATAC-seq improves GRN inferences. Consistent with other contexts (Argelaguet et al, 2022 Preprint; Kamimoto et al, 2023), we found that incorporating chromatin accessibility/TF motif analysis together with transcriptional inferences improves the accuracy of GRN inference, by first narrowing down the choice of TF targets from which to infer the mRNA transcript co-expression network, a principle that is likely to be especially important with small sample sizes. Notably, we analyse RNA-seq data generated using SMART-seq2 and the sequencing method and depth of sequencing will likely impact on the choice of GRN inference method.

Here, we showed that incorporating complementary epigenomic data with transcriptomic data improves the reproducibility of inferred GRNs. Furthermore, it has enabled us to make predictions about GRNs operational in early human embryos that are consistent with an understanding of the function and association of the regulators in other developmental and stem cell contexts. We suggest that for network inference using advanced regression methods, it may be preferable to pre-filter using epigenomic data to first narrow down the TF targets in scRNA-seq datasets because this gives the network inference algorithm an

**Figure 4. TF-gene regulatory networks predicted by MICA to be regulated by, or to regulate, NANOG, GATA4, or CDX2.**
Target networks contain a maximum of 25 potential target TFs of the hub (or central) TF, whereas regulator networks include a maximum of 25 TFs that potentially regulate the hub. The average expression of the transcript is represented by the size of the TF node. MI scores are represented by the thickness of the line in the predicted network. To further refine the MICA predictions, we used the Spearman's rank correlation coefficient between the expression levels of the source and target nodes to define correlated or anti-correlated expression. Correlated or anti-correlated node pairs correspond to positive or negative Spearman's rank coefficient with $P$-value smaller than 0.1. Node pairs with a $P$-value equal to or larger than 0.1 were defined as uncorrelated, though they are predicted by MICA. **(A)** Network of TFs predicted to be targeted by NANOG. **(B)** Network of TFs that are predicted to regulate NANOG. **(C)** Network of TFs predicted to be targeted by GATA4. **(D)** Network of TFs that that are predicted to regulate GATA4. **(E)** Network TFs predicted to be targeted by CDX2. **(F)** Network of TFs that are predicted to regulate CDX2.

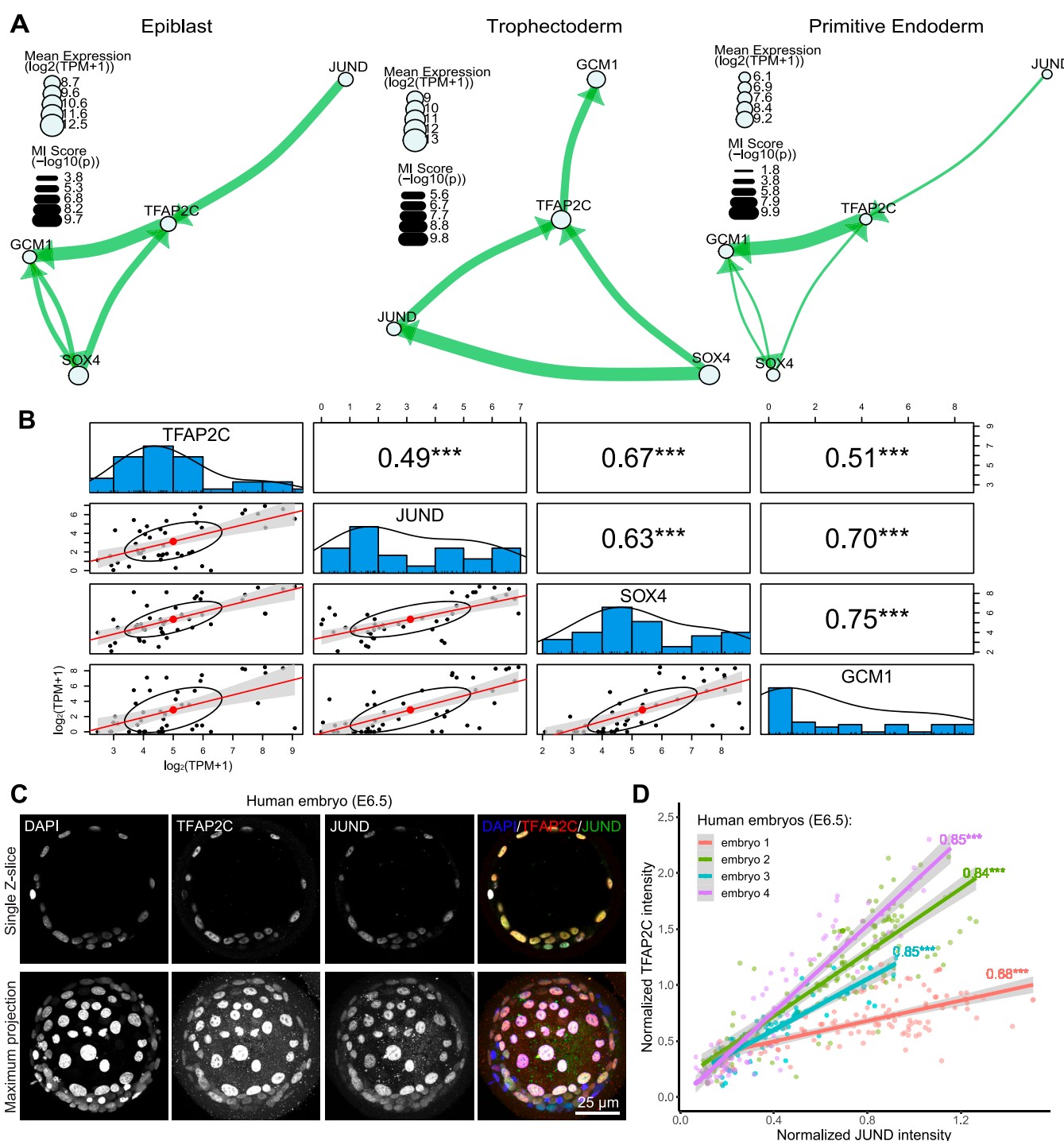

**Figure 5. MICA predicted conserved TFAP2C interactions in human early embryo EPI, TE, and PE cells.**
**(A)** Network composed of TFAP2C, JUND, SOX4, and GCM1 in EPI (left), TE (middle), and PE (right). **(B)** Correlation plots between TFAP2C, JUND, SOX4, and GCM1 in TE cells. Bottom left: log-transformed RNA expression of genes in single cells. Diagonal: distribution of log-transformed RNA expression for TFAP2C, JUND, SOX4, and GCM1. Top right: Spearman correlation between TF pairs (***P < 0.001). **(C)** Immunofluorescence staining of TFAP2C and JUND in E6.5 human blastocysts. **(D)** Correlation plot of TFAP2C and JUND protein expression intensity in nuclei of human E6.5 blastocysts. Numbers on the plot are Pearson correlation value (***P < 0.001).

easier time of selecting the regulators from transcriptome data, which is consistent with the experience of others using alternative GRN prediction methods (Badia-i-Mompel et al, 2023; Fleck et al, 2023; Kamimoto et al, 2023; Kartha et al, 2022). Alternative methods

incorporate epigenomic data after inference of the mRNA transcript co-expression network (González-Blas et al, 2022 *Preprint*) and this has been applied successfully to infer GRNs in the fly brain (Janssens et al, 2022). It will be interesting to determine how

the type and sequential order of incorporating multi-omics datasets impact on GRN predictions.

We also note some limitations on the interpretation of the GRNs predicted. For example, one ATAC-seq peak could cover multiple transcription factor binding sites in a region. In such cases, we include all TFs with motifs mapping to this region as potential regulators of the gene with the closest TSS to the region, for subsequent transcript co-expression network modelling. However, it will be interesting to determine if this can be refined, by applying the recently developed chromVAR-Multiome method to human blastocysts to generate an in silico ChIP-seq library for this context (Argelaguet et al, 2022 Preprint). Moreover, the ATAC-seq data applied in this study analysed the ICM data in bulk, without distinguishing between EPI and PE cells. Therefore, cell type–specific chromatin accessibility could not be considered, and specific interactions may have been missed because of the heterogeneity of the data or if the ICM CA data failed to reflect developmentally cis-regulations of more developmentally progressed EPI and PE cells. In addition, predictions based on the nearest TSS will miss long-range enhancers that are known to be important for gene regulation (Schoenfelder & Fraser, 2019). In the future, integration of single-cell or nuclei-matched transcriptome and ATAC-seq chromatin accessibility data, like recent studies in the mouse (Argelaguet et al, 2022 Preprint), would be preferable to apply in the human blastocyst context. We note that the GRN interference methods predicted edges that overlap between the four inference methods. Identifying overlapping inferences by comparing more than one GRN inference method may be a strategy to identify network edges with more confidence because of the agreement between several inference methodologies. However, this strategy may also miss some edges which can only be detected by one method and not another.

We also note that so far, we have separately modelled GRN structure to specific preimplantation embryonic cell types constrained to a single developmental time-point. In the future, we seek to model dynamic GRN structure in transcriptionally distinct human blastocyst lineages. Moreover, we seek to integrate CUT&RUN or CUT&Tag TF-DNA binding analysis (Skene & Henikoff, 2017; Kaya-Okur et al, 2019; Meers et al, 2019) for key putative developmental regulators, such as GATA3, to further narrow down experimentally validated occupancy from the ATAC-seq predictions we used in this study, similarly to a recent application in mouse blastocysts (Hainer et al, 2019; Hayashi & Inoue, 2023), though we note that in this context, TF occupancy studies will be restricted to a few TFs with good quality antibodies. It will also be important to determine which cis-regulatory regions are required for target gene regulation through systematic perturbation studies. As the topic of GRN inference is currently receiving much attention from the computational biology community, it will also be important in subsequent work to compare our pipeline with the latest alternatives beyond SCENIC, such as scMTNI (Zhang et al, 2023).

In summary, the MICA network analysis pipeline we developed is a tool that can be applied to challenging-to-study developmental contexts with limited sample size, such as the human blastocyst, to make predictions about TF interactions that can be experimentally tested in the future. As more datasets become available, we anticipate that the networks predicted will be further refined.

# Materials and Methods

## Ethics statement

All human embryo experiments followed all relevant institutional and national guidelines and regulations.

This study was approved by the UK Human Fertilisation and Embryology Authority (HFEA): research licence number R0162, and the Health Research Authority's Research Ethics Committee (Cambridge Central reference number 19/EE/0297).

The process of licence approval entailed independent peer review along with consideration by the HFEA Licence and Executive Committees. Our research is compliant with the HFEA code of practice and has undergone inspections by the HFEA because of the licence was granted. Research donors were recruited from patients at Bourn Hall Clinic, Homerton University Hospital, The Bridge Centre and IVF Hammersmith.

Informed consent was obtained from all couples who donated surplus embryos after IVF treatment. Before giving consent, people donating embryos were provided with all of the necessary information about the research project, an opportunity to receive counselling and the conditions that apply within the licence and the HFEA code of practice. Donors were informed that, in the experiments, embryo development would be stopped before 14 d post-fertilization, and that subsequent biochemical and genetic studies would be performed. Informed consent was also obtained from donors for all the results of these studies to be published in scientific journals. No financial inducements were offered for donation. Consent was not obtained to perform genetic tests on patients and no such tests were performed. The patient information sheets and consent document provided to patients are publicly available (https://www.crick.ac.uk/research/a-z-researchers/researchers-k-o/kathy-niakan/hfea-licence/). Donated embryos surplus to the IVF treatment of the patient were cryopreserved and were transferred to the Francis Crick Institute where they were thawed and used in the research project.

## ATAC-seq data processing and analysis

Chromatin accessibility profiles from the ICM and TE were obtained from the data produced by Liu et al with the LiCAT-seq protocol (Liu et al, 2019). Alignment to the reference genome (GRCh38), peak calling, and annotation were performed with nf-core/atacseq v1.1.0 (Ewels et al, 2020). Then, we carried out footprinting followed by TF motif enrichment analysis in the regions of open chromatin using rgt-hint v0.13.0 with TF binding models from HOCOMOCO and JASPAR (Li et al, 2019). Finally, we associated the TFs that exhibited over-represented motifs in these regions with their closest transcription-starting sites (most distances ranging from −5 to 10 kbp).

## scRNA-seq data processing and analysis

We integrated scRNA-seq data from three different studies (Yan et al, 2013; Blakeley et al, 2015; Petropoulos et al, 2016) focussing on the three cell types present at the late blastocyst stage (Figs 1 and S3A). Cell type annotations were taken from the work by Stirparo et al

(2018). Alignment to the reference genome (GRCh38) and calculation of gene counts and TPM-normalised counts were performed on each dataset separately with nf-core/rnaseq v1.4.2 (Ewels et al, 2020). The resulting gene expression matrices were integrated and normalised (log-counts and batch-corrected counts) using Bioconductor tools (Amezquita et al, 2020). The final set of cells was manually curated based on the UMAP representation of the batch-corrected data (see Fig S3 and Table S3). The list of 25,098 genes in the expression matrices was reduced to the unique set of TFs and TSS derived from the motif enrichment analysis applied to the ATAC-seq data. We used the ICM TFs and TSS for the EPI and PE matrices (12,780 genes) and the TE TFs and TSS for the TE matrices (12,981 genes).

### Network inference methods

For network inference, we compared the best performing strategy in the DREAM5 challenge, GENIE3 (Huynh-Thu et al, 2010), with a non-linear alternative based on MI (Faith et al, 2007), the Spearman's rank correlation coefficient, and the L0L2 sparse regression method that we applied using recent advances in sparse multivariate statistical modelling (Hazimeh & Mazumder, 2018 Preprint; Bartlett et al, 2019 Preprint) as follows.

For GENIE3, transcription factors are ranked according to the degree of variability in their expression level and how the expression of the putative regulator correlates with a target gene. This ranking is then used to construct the co-expression network for all genes and transcription factors, by thresholding the algorithm's variance reduction score at the 10th percentile of its empirical distribution. The GRN is then inferred as the intersection of the edges in this co-expression network with the edges in the network of all possible gene regulations derived from the chromatin accessibility/DNA binding motif data. This network of all possible gene regulations is defined as all network edges from regulating TF to regulated gene, where an edge represents a DNA binding motif for the TF in regulatory DNA within open chromatin in the regulated gene. For GENIE3, all default settings were used.

With the Spearman correlation coefficient, a weighted co-expression network is inferred as the absolute value of the correlation coefficient. The GRN is then inferred as the intersection of the edges in this co-expression network with the edges in the network of all potential gene regulations derived from the chromatin accessibility/DNA binding motif data.

For L0L2 regression (described in more detail below), the model automatically chooses a ranked subset of TFs from those predicted as regulators of the target gene in the chromatin accessibility/DNA binding motif data inferences. In this way, each target gene takes its turn for a model to be fitted around that target gene. After the model has been fitted to every target gene, the global GRN can be constructed by combining the local networks fitted around each target gene. For L0L2 regression, sparsity hyperparameters were chosen using the L0Learn package's internal cross-validation.

In more detail, for L0L2 regression, we start with a linear model of the expression level $y$ of the regulated target gene (Dobra et al, 2004), in terms of the expression levels $x_1, x_2, ..., x_p$, of $p$ transcription factors. We want to use the size of the fitted model coefficients $b_1, b_2$, etc. to measure the strength of regulation of the target gene by transcription

factors (TFs) 1, 2, etc. We use sparse regression to find $b_1$, $b_2$, etc., as this specifically minimises the number of non-zero model coefficients, by requiring that the coefficients $b_j$ are set to zero as much as possible. This leads to a more parsimonious model, in which a relatively small number $p' \ll p$ of transcription factors is inferred to be regulating the target gene, as a result of non-zero. Sparse regression minimizes:

$$\left[ y - \left( a + b_1 x_1 + b_2 x_2 + ... + b_p x_p \right) \right]^2 + \psi \tag{1}$$

where $\psi$ "penalises" models with values of $b_j$, ($j = 1, ..., p$) further from zero. The most popular choices for $\psi$ include $\psi = \lambda \Sigma b_j^2$, which is called "ridge regression" (L2 regression), and $\psi = \lambda \Sigma |b_j|$, which is called "the lasso" (L1 regression). It can be shown that Equation (1) is equivalent to applying the constraint $\Sigma b_j^2 < t^2$ for $\psi = \lambda \Sigma b_j^2$ (L2 regression) or $\Sigma |b_j| < t$ for $\psi = \lambda \Sigma |b_j|$ (L1 regression). Then, "best-subset regression" (L0 regression) specifies the number of transcription factors that can have non-zero $b_j$. It does this by using the constraint $\Sigma I(|b_j| > 0) \leq k$, which is equivalent to $\psi = \lambda \Sigma I(|b_j| > 0)$ in Equation (1). Combinations of these constraints are often more effective, such as L1L2 regression (also called "the elastic net" [Zou & Hastie, 2005], which has proven very successful in genomics), with penalty term $\psi = \gamma \Sigma |b_j| + \lambda \Sigma b_j^2$. Recently, L0L2 regression has been proposed as an improvement (Hazimeh & Mazumder, 2018 Preprint), with penalty term $\psi = \gamma \Sigma I(|b_j| > 0) + \lambda \Sigma b_j^2$. We use L0L2 regression in this study, for reasons as follows. Sparse regression using the L0 penalty is an ideal model for inferring a minimal set of regulating transcription factors, because it specifically selects the best set of $k$ transcription factors for the model (i.e., the "best subset," of the available transcription factors). Combining with the L2 penalty in L0L2 makes the model better specified for the data, by minimizing the total size of the linear model coefficients so that the most important TFs receive the largest linear model coefficients. We use sparse L0L2 regression to infer the best subset of regulators of each target gene, from the full list of TF-gene associations supported by the chromatin accessibility data.

For MI-based inference, the co-expression network is estimated first, as follows. We use an empirical estimate of the distribution of the MI values for each gene (Daub et al, 2004; Faith et al, 2007). Writing the estimated MI between the expression levels of genes $x$ and $y$ as $M_{xy}$, we estimate the (assumed) standard Gaussian-distributed variable $z_{xy} \sim N(0,1)$ according to the equation:

$$z_{xy}^2 = F^{-1} \left[ \hat{F}_{y|x}(M_{xy}) \right]^2 + F^{-1} \left[ \hat{F}_{x|y}(M_{xy}) \right]^2 \tag{2}$$

where $F$ is the $N(0,1)$ cumulative distribution function (c.d.f.), and $\hat{F}_{y|x}$ and $\hat{F}_{x|y}$ are the empirical c.d.f.s of $M_{xy}$ conditioned on $x$ and $y$, respectively. For each of these (assumed) $N(0,1)$ distributed $z_{xy}$ we calculate $P$-values, and then threshold each of these at a false discovery rate of 5%. Again, the GRN is inferred as the intersection of the edges in this co-expression network with the edges in the network of all possible gene regulations derived from the chromatin accessibility/DNA binding motif data.

We also considered a "random predictor" that outputs a random ordering of all the possible TF-gene interactions.

The putative regulatory links predicted by each of these methods using the scRNA-seq data were evaluated as is but also subjected to

a filtering process in which only TF-gene associations supported by the chromatin accessibility data were considered in the final network (for GENIE3, MI, and Spearman correlation methods). In the case of L0L2 sparse regression, only TF-gene associations supported by the chromatin accessibility data were considered as potential regulators of the target gene in the regression model. To identify these refined predictions, we added the *+CA* suffix to the name of the GRN inference methods. In both cases, to generate the final network, we selected the top 100,000 edges by ranking edges according to their network score. We define network score as the absolute correlation coefficient or linear regression coefficient (in the case of Spearman's correlation and sparse regression respectively), or as $-\log_{10}p$ for mutual information, or the GENIE3 score.

## Simulation study

For our simulation study, we used the dagitty package in R to generate synthetic gene expression data based on pre-defined GRNs with pre-determined network structure. These GRNs were generated with network edge density of $\rho$ = 0.07, i.e., 7% of all possible edges (or gene regulations), are present in the network (this value was estimated from the available ATAC-seq+TF motif data). The synthetic datasets were generated with a range of sample size $n$ and number of potential regulators of each node (gene) $p$, using linear models, as follows. For each combination of $n$ and $p$, we generated 100 GRNs; then corresponding observed gene expression datasets were generated for each of the GRNs by specifying that the expression level of a downstream gene should depend on a linear combination of the expression levels of its upstream regulators:

$$x_j = \sum_{j' \neq j} b_{j'} x_{j'} + e \qquad (3)$$

where $x_j$ is the expression level of the regulated (downstream) gene, $x_{j'}$ are the expression levels of the $p$-1 regulating (upstream) TFs, $b_{j'}$ are corresponding linear combination weightings, and $e$ is an error term. The different network inference methods (GENIE3, L0L2 sparse regression, MI, Spearman correlation) were then applied to these generated data, and the results were compared with the known pre-determined network structure. This comparison was assessed by the AUC statistic, the "area under the ROC (receiver-operator characteristic) curve."

## Statistical evaluation metrics

We calculated a reproducibility score $R$ as a bootstrap estimate of the posterior probability of observing an edge $E$ given the dataset $D$, $R = P(E|D)$ (Pe'er, 2005), for the top 100,000 predicted edges. We also carried out twofold cross-validation in two ways as follows. We randomly split each dataset (EPI, PE, and TE) in two groups with the same number of cells 10 times (repeated twofold cross-validation), applied the inference methods (with or without chromatin accessibility refinement), took the top 1% of all possible regulatory interactions from one group and quantified the extent to which the top interactions inferred from the other group coincide with that reference using the AUPRC. In addition, we calculated a normalised L2 loss comparing all the network inferences from the two network

fits on the two folds of the data (at the level of the network scores). The normalised L2 loss is then defined as the L2 loss comparing the network scores from each of the two data folds, divided by the product of the square roots of the L2 norms of the network scores of each of the two data folds.

## Biological evaluation metrics

For each combination of inference method and normalisation approach, we computed the intersection between the GRN edges predicted to identify interactions that are specific to the EPI, PE, and TE, as well as the interactions common to the three cell types. In set notation this corresponds to EPI-specific = (EPI\PE)\TE, PE specific = (PE\EPI)\TE, TE specific = (TE\EPI)\PE and Common = EPI ∩ PE ∩ TE. Then, we used the out-degree of NANOG, GATA4, and CDX2 (marker TFs of the EPI, PE, and TE, respectively) as a proxy for their activity in each one of the four networks. We considered that a marker was active if its normalised out-degree (i.e., the proportion of all genes it regulates in that GRN) was at least the median normalised out-degree across all networks.

We calculated a validation score $V$ as the number of relevant gene sets that were enriched at a significance level of 10% ($P < 0.1$) in a geneset enrichment analysis performed with the genes regulated by the 25 most connected TFs and that were part of the 500 interactions with the highest prediction scores produced by the GRN inference methods for the EPI and TE. The list of gene sets for the EPI and TE were manually curated and represent the most relevant pathways and biological processes in each one of these cell types (see Tables S1 and S2). The starting list of gene sets comprised all the pathways and Gene Ontology Biological Process (GO BP) terms collated by the Bader Lab at the University of Toronto on 01 October 2020 (http://download.baderlab.org/EM_Genesets/October_01_2020/Human/symbol/Human_GOBP_AllPathways_no_GO_iea_October_01_2020_symbol.gmt). To facilitate the comparison between $V$ scores from different GRN inference methods within a cell type, we normalised $V$ to the maximum score attained by a network predictor for that cell type (i.e., EPI or TE).

## Human embryo culture

Human embryos were cultured as previously described (Gerri et al, 2020). Vitrified human blastocyst stage embryos were thawed using Kitazato Thawing Media (VT602, order number 91121) following the manufacturer's instructions.

Human embryos were cultured in drops of pre-equilibrated Global medium (LGGG-20; LifeGlobal) supplemented with 10% human serum albumin (GHSA-125; LifeGlobal) and overlaid with mineral oil (ART-4008-5P; Origio). Preimplantation embryos were incubated at 37°C and 5% $CO_2$ in an EmbryoScope+ time-lapse incubator (Vitrolife) and cultured for up to 24 h for human blastocyst.

## Immunofluorescence staining

Embryos were fixed with freshly prepared 4% PFA in PBS. Fixation was performed for 20 min at RT for embryos. Embryos were then washed three times in 1 × PBS with 0.1% Tween-20 to remove residual PFA. Permeabilization was performed with 1 × PBS with 0.5% Tween-20 and followed by blocking in blocking solution (10%

donkey serum in 1 × PBS with 0.1% Tween-20) for 1 h at RT on a rotating shaker. Then, antibody incubation was performed with primary antibodies diluted in blocking solution overnight at 4°C on rotating shaker. The following day, embryos and cell cultures were washed in 1 × PBS with 0.1% Tween-20 for three times, and then incubated with secondary antibodies diluted in blocking solution for 1 h at RT on a rotating shaker in the dark. Next, embryos and cell cultures were washed in 1 × PBS with 0.1% Tween-20 for three times. Finally, embryos were placed in 1 × PBS with 0.1% Tween-20 with Vectashield and DAPI mounting medium (H-1200; Vector Lab) (1:30 dilution). Embryos were placed on μ-slide eight-well dishes (80826; Ibidi) for confocal imaging. The antibodies and concentrations used are reported in Table S4.

### Confocal imaging

Confocal immunofluorescence images were taken with a Leica SP8 confocal microscope. 2-μm thick optical sections were collected for embryo Z-stack imaging.

### Nuclei segmentation and quantification

Nuclei segmentation and quantification were performed on Z-stack confocal images taken at 2-μm thick optical sections. Stardist Weigert et al (2020) was used for nuclei segmentation followed by CellProfiler (Stirling et al, 2021) for nuclei tracking and fluorescence intensity quantification based on a customized pipeline modified from Lea et al (2021). First, LIF format confocal images were exported into multi-channel Z-stack TIF format for each Z-stack image using ImageJ (Schneider et al, 2012). Then, Stardist was used to identify the nuclei based on the DAPI channel (358 nm). Finally, Stardist outputs were split into single-channel single-stack images and loaded into the CellProfiler v4.2.5 to track nuclei across image slices and quantify the fluorescence intensity. Customized pipeline and scripts can be found here: https://github.com/galanisl/early_hs_embryo_GRNs.

# Data Availability

scRNA-seq and ATAC-seq data were obtained from the European Nucleotide Archive (accession numbers: PRJNA153427, PRJNA277181, PRJEB11202, and PRJNA494280). Data pre-processing and analysis scripts are available at https://github.com/galanisl/early_hs_embryo_GRNs.

# Supplementary Information

# Acknowledgements

We thank Zimeng Chen for contributing to the initial analysis of the synthetic datasets. We thank members of the KK Niakan laboratory for helpful discussions and feedback on the manuscript. Work in the laboratory of KK Niakan was supported by the Wellcome (221856/Z/20/Z) and the Wellcome Human Developmental Biology Initiative (215116/Z/18/Z). Work in the laboratory of KK Niakan was also supported by the Francis Crick Institute, which receives its core funding from Cancer Research UK (CC2074), the UK Medical Research Council (CC2074), and the Wellcome Trust (CC2074). The work carried out by TE Bartlett was partly supported by MRC grant (MR/P014070/1). For the purpose of open access, the authors have applied a CC BY public copyright licence to any author-accepted manuscript version arising from this submission.

# Author Contributions

G Alanis-Lobato: conceptualization, data curation, software, formal analysis, supervision, validation, investigation, visualization, methodology, and writing—original draft, review, and editing.
TE Bartlett: conceptualization, resources, data curation, software, formal analysis, supervision, funding acquisition, validation, investigation, visualization, methodology, project administration, and writing—original draft, review, and editing.
Q Huang: conceptualization, formal analysis, validation, investigation, visualization, methodology, and writing—original draft, review, and editing.
CS Simon: supervision, methodology, and writing—review and editing.
A McCarthy: investigation.
K Elder, P Snell, and L Christie: resources.
KK Niakan: conceptualization, resources, supervision, funding acquisition, project administration, and writing—original draft, review, and editing.

# Conflict of Interest Statement

The authors declare that they have no conflict of interest.

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
