## [Reviewer comments · Life Science Alliance]

MICA: A multi-omics method to predict gene regulatory networks in early human embryos

Gregorio Alanis-Lobato, Thomas Bartlett, Qiulin Huang, Claire Simon, Afshan McCarthy, Kay Elder, Phil Snell, Leila Christie, and Kathy Niakan

DOI: <https://doi.org/10.26508/lsa.202302415>

Corresponding author(s): Kathy Niakan, University of Cambridge and Gregorio Alanis-Lobato, Boehringer Ingelheim

Review Timeline:	Submission Date:	2023-10-04
	Editorial Decision:	2023-10-05
	Revision Received:	2023-10-06
	Editorial Decision:	2023-10-10
	Revision Received:	2023-10-12
	Accepted:	2023-10-13

Transaction Report:

Please note that the manuscript was reviewed at Review Commons and these reports were taken into account in the decision-making process at *Life Science Alliance*.

Reviews

Review #1

****Summary****

The authors proposed MICA strategy as an attempt to infer gene regulatory network at the blastocyst stage of early embryo development which features limited sample size. While the motivation seems reasonable to me and the results showed several interesting insights, the methodological novelty and significance of this study need further elaboration, and the evaluation/benchmark part is largely insufficient.

****Major comments****

- The proposed strategy (i.e. combining gene expression-based regulatory inference with cis-regulatory evidence) have been well developed (and implemented) by multiple published works like SCENIC and CellOracle, which is also properly acknowledged by the authors in the discussion section too. This leads to a serious concern on the major methodological contribution of this work.
- Most of the compared network reconstruction methods involve hyper-parameters setup (e.g., sparsity regularization weights of the regression methods). The authors did not discuss how these hyper-parameters were chosen.
- For the real-world blastocyst data, the network prediction methods were compared in terms of their reproducibility across validation folds (Fig. 3, Fig. S4-6). However, reproducibility does not necessarily imply accuracy. In fact, statistical learning methods are generally subject to the bias-variance tradeoff, where lower variance (i.e., higher reproducibility) could imply higher bias in model prediction. While there is a lack of gold-standard ground truth to evaluate network accuracy in real biological systems, silver-standards like the ranking of known regulatory interactions in the predictions could be employed as an indirect estimate.
- The gene set enrichment results were reported only on EPI and TE cell types (Fig. 4C and Fig. S12), due to the reason that CA data is only available for TE and ICM. However, many of the other results presented in Fig. 3-6 did include the PE cell type albeit using the same CA data. It is not particularly convincing why the cell type inclusion standard for gene set enrichment is different from the other results.
- The authors cited TF binding in cis-regulatory regions as supporting evidence of several MICA-inferred regulatory interactions (e.g., NANOG -> ZNF343). However, the same cis-regulatory evidence has already been used in the CA filtering step. All interactions passing CA filtering should in principle have TF-binding support. It would be more convincing if the authors provided other types of evidence as independent support, such as genetic associations like eQTL, experimental perturbations like gene knockdown/knockout, etc.
- Many of the MICA-inferred regulatory interactions do not exhibit Spearman correlation (Fig. 5, Fig. S17), which could probably be explained by the ability of mutual information to capture complex non-monotonic dependencies. It would be interesting to provide further investigation on these "uncorrelated" edges, which may help demonstrate the superiority of mutual information over Spearman correlation.
- The authors conducted immunostaining experiments to validate the MICA-inferred regulatory interaction between TFAP2C and JUND. While the identified protein co-localization is a step further than RNA co-expression, it is still correlation rather than causality. Additional evidence like the effect of knockout/knockdown perturbations would be more convincing.

****Minor comments****

- The γ symbols in AP-2 γ are not correctly rendered.
- The UMAP figures (Fig. 4A, Fig. S7) are of low resolution compared to other figures.
- As the authors are focused on studying the blastocyst regulatory network, the inferred regulatory interactions should be provided as supplementary data.

Given the concerns listed above, I still hold doubts on the significance of the manuscript in its current form. In particular, the major contribution of this work, in methodological senses, seems to be the specific choice of mutual information for regulatory inference in the low-data regime, which may have a limited audience and impact.

Review #2

In this work, Alanis-Lobato et al apply different GRN inference methods on scRNA-seq data from human blastocysts. By integrating the data with ATACseq, they manage to address the small sample size challenge and predict novel TF-gene interactions that they later validate with immunofluorescence. Main take-home-messages from this work are that proper

GRN inference methods work better upon integration of different omic technologies (here RNA and ATAC seq) and proper data normalisation strategies (logTPM or logFPKM).

Hereby I present some minor concerns and questions that I have after reading the manuscript, that I hope the authors can address.

- In the abstract, it would be adequate to already mention which normalisation method works the best.
- In Fig. 1:
 - Describe what are squares and circles
 - In the GRNs refined by keeping CA-predicted regulations only, mention that this are Cis interactions
 - The ATAC seq shows KRT8, GATA3, RELB motifs, while the rest of the figure is very general. Maybe make the ATAC-seq peaks panel also as a sketch and relate it to the square/circles graphs on the right hand side to showcase how the filtering of the network is performed.
 - The caption says Five GRN inference approaches, while abstract and text say 4. It is clear after reading that the 5th is a random approach. However, it was a surprise at first.
 - How the Simulation study was performed is not understandable for non experts as it is described in the Methods section. This is an important approach in general, and I think the audience would benefit if the authors add a full section about it in their supplementary data.
- Fig. 2:
 - As it is, it is hard to tell the difference between GRN inference methods for a given sample size and number of regulators. Could the authors add a comparative panel for this (maybe some scatter plots would be enough)? MI by itself looks worse here?
 - When mentioning "samples" (e.g. last paragraph of section 1 in results), do the authors refer to "cells"?
 - What about normalisation effects in the simulated data?
 - Figure S7 should be cited in the first paragraph of section 2 in results. Could the authors add a panel to indicate whether the data is SMART-seq2 or 10X.
 - In the association of inferred GRNs to human blastocyst cell lineages, the authors find the GRN edges predicted that overlap between the 4 inference methods in each cell type. Do they, therefore, recommend to always use more than one GRN inference method?
 - If the CA data used was only generated for the TE and ICM only, how do the authors use it to perform MICA on PE?

In this paper, one main message is that to infer GRN one should combine different omic datasets. This does not come as a surprise and has been published before. What it is very well addressed in this study is the problem of the sample size: the authors decide to test GRN inference methods in the human blastocyst, for which currently we do not have a lot of sequencing data available. Interestingly, they find that 1k cells should be enough to infer relevant GRN. Maybe the manuscript would benefit if the authors emphasize this more in their text.

Interestingly, and despite the fact that the sample size here is below 1k, the authors identify novel regulatory relationships between TFs for different cell types, that they also validate.

This paper will be relevant to a wide audience of scientists interested in human developmental biology, or in the development of computational approaches to analyse single cell sequencing data.

October 5, 2023

Re: Life Science Alliance manuscript #LSA-2023-02415-T

Kathy K Niakan
University of Cambridge
United Kingdom

Dear Dr. Niakan,

Thank you for submitting your manuscript entitled "MICA: A multi-omics method to predict gene regulatory networks in early human embryos" to Life Science Alliance. We invite you to re-submit the manuscript, revised according to your Revision Plan.

Thank you for this interesting contribution to Life Science Alliance. We are looking forward to receiving your revised manuscript.

Sincerely,

Eric Sawey, PhD
Executive Editor
Life Science Alliance
<http://www.lsa-journal.org>

B. MANUSCRIPT ORGANIZATION AND FORMATTING:

*****IMPORTANT:** It is Life Science Alliance policy that if requested, original data images must be made available. Failure to provide original images upon request will result in unavoidable delays in publication. Please ensure that you have access to all original microscopy and blot data images before submitting your revision. *******

1. General Statements [optional]

We are grateful to both reviewers for reviewing our manuscript, and for providing very helpful feedback as to how we can improve this work. We have now implemented nearly all of the changes as recommended, and provide responses to these points below.

In terms of novelty, while recent pre-prints and publications have suggested that the application of multi-omics analysis improves GRN inference, there has yet to be a systematic comparison of linear and non-linear machine learning methods for GRN prediction from single cell multi-omic data. There are many computational and statistical challenges to such a study, and we therefore believe that others in the field will be especially interested in our systematic comparison of network inference methods, especially given the increased interest and utility of multi-omic data.

In addition, we report the first comprehensive inference of GRNs in early human embryo development. This is a particularly challenging to study developmental context given genetic variation, limitations of sample size due to the precious nature of the material and regulatory constraints. We anticipate that the methodology we developed and datasets we generated will be informative for computational, developmental and stem cell biologists.

We have uploaded all the network predictions on FigShare and these can be accessed using the following link: <https://doi.org/10.6084/m9.figshare.21968813>. In addition, we anticipate that the computational and statistical codes and pipelines we developed (available on https://github.com/galanisl/early_hs_embryo_GRNs) will be applied to other cellular and

developmental contexts, especially in challenging contexts such as human development, non-typical model organisms and in clinically relevant samples.

Insert here a point-by-point reply that explains what revisions, additional experimentations and analyses are planned to address the points raised by the referees.

Please insert a point-by-point reply describing the revisions that were already carried out and included in the transferred manuscript. If no revisions have been carried out yet, please leave this section empty.

Reviewer 1

Major comments

- The proposed strategy (i.e. combining gene expression-based regulatory inference with cis-regulatory evidence) have been well developed (and implemented) by multiple published works like SCENIC and CellOracle, which is also properly acknowledged by the authors in the discussion section too. This leads to a serious concern on the major methodological contribution of this work.

We would like to note that our study is the first to comprehensively evaluate machine learning linear or non-linear gene regulatory network prediction strategies from single cell datasets combined with available multi-omic data. We also apply these methods to the challenging to study context of human early embryogenesis. There are specific methodological challenges arising in this context that other published work has not yet addressed. In particular, the precious nature of the source material means that sample sizes are limited, unlike the contexts where SCENIC and CellOracle were applied. Notably, the numbers of cells available for downstream analysis is typically several orders of magnitude fewer than when scRNA-seq data are collected from adult human tissue or from cell culture. This restriction on sample sizes places corresponding restrictions on statistical power, and is therefore likely to mean that different statistical network inference methodology is optimal in specific contexts. Furthermore, the inclusion of multi-omic data from complementary platforms (such as ATAC-seq data) becomes even more important in this context to mitigate the effect of reduced sample sizes. These issues are very important for choice of gene regulatory network inference methodology in relation to studies of the human embryo development, and ours is the first study to address

these issues directly in any context. We have further clarified the novelty of our work in the manuscript in the abstract, introduction and discussion sections.

- Most of the compared network reconstruction methods involve hyper-parameters setup (e.g., sparsity regularization weights of the regression methods). The authors did not discuss how these hyper-parameters were chosen.

For sparse regression, the hyperparameter controlling sparsity was set by cross-validation (CV), using the internal CV function of the R package. All default settings for GENIE3 were used. This information has now been added to the manuscript (in the Methods section), along with a description of the implementation of the mutual information method we use.

- For the real-world blastocyst data, the network prediction methods were compared in terms of their reproducibility across validation folds (Fig. 3, Fig. S4-6). However, reproducibility does not necessarily imply accuracy. In fact, statistical learning methods are generally subject to the bias-variance tradeoff, where lower variance (i.e., higher reproducibility) could imply higher bias in model prediction. While there is a lack of gold-standard ground truth to evaluate network accuracy in real biological systems, silver-standards like the ranking of known regulatory interactions in the predictions could be employed as an indirect estimate.

We thank the reviewer for the opportunity to clarify this point. We would like to avoid any misunderstanding of the reproducibility statistic R , as follows. A higher value of R indicates that the fitted model would generalise well to new data; i.e., $R=1$ indicates that the model is robust (stable) to perturbations of the data-set. We note that this is not the same as analysing the residual variance of the data after model fitting and related over-fitting (i.e., bias-variance trade-off). The variance that is referred to when discussing bias-variance trade-off is the mean-squared error (of data compared to model), which is not the same as what is assessed by reproducibility statistic R . Specifically, R is a Bayesian estimate of the posterior probability of observing a gene regulation given the data. R is calculated by taking a random sample of the data, doing the network inference again, checking if each gene regulation still appears in the GRN, and then recording (as the R statistic) the average fraction of inclusions over many repetitions. So when we have R close to 1, this indicates that our model predictions generalise well to new data, which is the opposite of what is suggested in this comment. In summary, the accuracy quantified by the reproducibility statistic R relates to the stability of the model predictions to perturbation of the data. We thank the reviewer for the helpful comment to draw our attention to this point, and have now clarified this point in the manuscript on page 5 line 223.

- The gene set enrichment results were reported only on EPI and TE cell types (Fig. 4C and Fig. S12), due to the reason that CA data is only available for TE and ICM. However, many of the other results presented in Fig. 3-6 did include the PE cell type albeit using the same CA data. It is not particularly convincing why the cell type inclusion standard for gene set enrichment is different from the other results.

We thank the reviewer for noting this and would like to clarify that we restricted the analysis to the EPI and TE, because similar lists of gene-sets were not available for primitive endoderm, where it is currently unclear which pathways are most relevant to this cell type. This has now been clarified in the manuscript on page 6, line 296.

- The authors cited TF binding in cis-regulatory regions as supporting evidence of several MICA-inferred regulatory interactions (e.g., NANOG -> ZNF343). However, the same cis-regulatory evidence has already been used in the CA filtering step. All interactions passing CA filtering should in principle have TF-binding support. It would be more convincing if the authors provided other types of evidence as independent support, such as genetic associations like eQTL, experimental perturbations like gene knockdown/knockout, etc.

We appreciate the reviewer's point. We address this by describing published ChIP-seq validation in human pluripotent stem cells which is widely used as a proxy for the study of the epiblast. We feel that the ChIP-seq validation in this context is an appropriate independent validation to support the MICA-inferred cis regulatory interactions predicted from the human embryo datasets we analysed. Our inferences from ATAC-seq data cannot identify TF-DNA binding directly. ChIP-seq data is a widely accepted independent methods to support the inferred interactions from ATAC-seq data.

We agree that knockdown/knockout would provide further evidence suggesting gene regulation, and indeed these are experiments we would like to conduct systematically in the future, but such perturbations are difficult to achieve at genome-wide scale, especially with very restricted quantities of human embryo material. Notably, these studies would not be evidence of direct regulation and the gold-standard in our opinion is to perturb the cis regulatory region to demonstrate its functional importance in gene regulation. These are important experiments to conduct systematically in the future. We also note that assessing quantitative trait loci in the context of human pre-implantation embryos is extremely challenging due to the restricted sample sizes and genetic variance in the samples collected.

- Many of the MICA-inferred regulatory interactions do not exhibit Spearman correlation (Fig. 5, Fig. S17), which could probably be explained by the ability of mutual information to capture complex non-monotonic dependencies. It would be interesting to provide further investigation on these "uncorrelated" edges, which may help demonstrate the superiority of mutual information over Spearman correlation.

We very much appreciate the reviewer's insightful comments. We have expanded on this in the manuscript on page 8 from line 420-430 and included Fig.S12 and S13B which are consistent with this point.

- The authors conducted immunostaining experiments to validate the MICA-inferred regulatory

interaction between TFAP2C and JUND. While the identified protein co-localization is a step further than RNA co-expression, it is still correlation rather than causality. Additional evidence like the effect of knockout/knockdown perturbations would be more convincing.

We agree that experimental perturbations would be informative. However due to the complexity of such an investigation in human embryos, it is beyond the scope of the current study. This is a future direction and we have clarified this in the discussion section.

Minor comments

- The γ symbols in AP-2 γ are not correctly rendered.

We note that this applies only to the way AP-2 γ appears on the Review Commons website, and we are trying to fix this issue. We hope this transformation after the manuscript upload will not apply to a subsequent transfer to a journal.

- The UMAP figures (Fig. 4A, Fig. S7) are of low resolution compared to other figures.

We thank the reviewer for noting this. These figures have now been added as vector graphics files to overcome this issue.

- As the authors are focused on studying the blastocyst regulatory network, the inferred regulatory interactions should be provided as supplementary data.

We have included all of the inferred gene regulatory interactions as a supplementary folder for the MICA predictions using FigShare: doi.org/10.6084/m9.figshare.21968813. We have included code to reproduce the inferred gene regulatory interactions for the other methods which we compared to MICA. Because this includes 100,000 regulatory interactions per method, we feel that it would be impractical to include the alternative inferred interaction as supplementary data.

Reviewer 2

Minor comments

- In the abstract, it would be adequate to already mention which normalisation method works the best.

This has now been added to the abstract and we appreciate this suggestion.

- In Fig. 1:

** Describe what are squares and circles*

This information has been included in the figure 1 legend.

** In the GRNs refined by keeping CA-predicted regulations only, mention that this are Cis interactions*

We have modified the figure 1 legend and the text on page 4, line 198, and throughout the manuscript, to clarify that these are putative cis-regulatory interactions.

** The ATAC seq shows KRT8, GATA3, RELB motifs, while the rest of the figure is very general. Maybe make the ATAC-seq peaks panel also as a sketch and relate it to the square/circles graphs on the right hand side to showcase how the filtering of the network is performed.*

We appreciate this suggestion and modified figure 1 accordingly.

** The caption says Five GRN inference approaches, while abstract and text say 4. It is clear after reading that the 5th is a random approach. However, it was a surprise at first.*

We have modified the figure 1 legend to clarify that we also compared random prediction in addition to the 4 GRN inference approaches.

- How the Simulation study was performed is not understandable for non experts as it is described in the Methods section. This is an important approach in general, and I think the audience would benefit if the authors add a full section about it in their supplementary data.

Further details have now been added to the subsection 'simulation study' in the Methods section.

- Fig. 2:

** As it is, it is hard to tell the difference between GRN inference methods for a given sample size and number of regulators. Could the authors add a comparative panel for this (maybe some scatter plots would be enough)? MI by itself looks worse here?*

We thank the reviewer for this helpful suggestion. This comparative plot has now been included in figure 1C and indicates that MI is on par with the other GRN inference methods using simulation RNA-seq data.

- When mentioning "samples" (e.g. last paragraph of section 1 in results), do the authors refer to "cells"?

We appreciate the reviewer pointing this out and have amended the text throughout to state that these are cells.

- What about normalisation effects in the simulated data?

With regards to the simulated data, normalisation effects are not relevant as we are generating data that are idealised and therefore not subject to unwanted sources of variation such as read depth. However, in future work, this could be investigated with an expanded simulation study and we appreciate the reviewer's suggestion.

- Figure S7 should be cited in the first paragraph of section 2 in results.

This has now been cited.

Could the authors add a panel to indicate whether the data is SMART-seq2 or 10X.

We thank the reviewer for the suggestion to clarify this, which we think is an important point. We have included a statement that all data used was generated using the SMART-seq2 sequencing technique in the figure legend. The choice of sequencing method/depth of sequencing will likely impact on the choice of GRN inference method and we have also clarified this in the discussion section on page 9, line 463.

- In the association of inferred GRNs to human blastocyst cell lineages, the authors find the GRN edges predicted that overlap between the 4 inference methods in each cell type. Do they, therefore, recommend to always use more than one GRN inference method?

Identifying overlapping inferences by comparing more than one GRN inference method may be a strategy to identify network edges with more confidence due to the agreement between several inference methodologies. However, this strategy may also miss some edges which can only be detected by one method and not another. We have included a statement in the discussion section to clarify this point on page 10, line 499.

- If the CA data used was only generated for the TE and ICM only, how do the authors use it to perform MICA on PE?

We appreciate that this is confusing and have since revised the manuscript on page 4, line 197 to state that the inner cell mass (ICM), comprises EPI (epiblast) and PE (primitive endoderm) cells. It may be that we miss putative cis-regulatory interactions if the ICM CA data does not reflect developmentally progressed PE and EPI cells and we have noted this caveat in the discussion section.

4. Description of analyses that authors prefer not to carry out

Please include a point-by-point response explaining why some of the requested data or additional analyses might not be necessary or cannot be provided within the scope of a revision. This can be due to time or resource limitations or in case of disagreement about the necessity of such additional data given the scope of the study. Please leave empty if not applicable.

Reviewer 1

- The authors conducted immunostaining experiments to validate the MICA-inferred regulatory interaction between TFAP2C and JUND. While the identified protein co-localization is a step further than RNA co-expression, it is still correlation rather than causality. Additional evidence like the effect of knockout/knockdown perturbations would be more convincing.

We agree with Reviewer 1 that experimental perturbations of TFAP2C and JUND to determine what consequence this has for interactions between these proteins would be informative. However due to the complexity of such an investigation in human embryos, we feel that this is beyond the scope of the current study. One option is to conduct the perturbations in human pluripotent stem cells, however it is unclear if the GRN in this context reflects the same interactions as human embryos and is a distinct question to address in the future. Moreover, while knockdown/knockout studies would be suggestive of up-stream regulation, it will not address the question of whether this is a direct or indirect effect without systematic further analysis including transcription factor-DNA binding (such as CUT&RUN, CUT&Tag or ChIP-seq) analysis as well as perturbations of the putative cis regulatory regions. These are all exciting future experiments and our study provides us and others with hypotheses to functionally test in the future. These are future directions and we have clarified this in the discussion section on page 10, line 504.

October 10, 2023

RE: Life Science Alliance Manuscript #LSA-2023-02415-TR

Prof. Kathy K Niakan
University of Cambridge
Centre for Trophoblast Research
Department of Physiology, Development and Neuroscience
Cambridge CB2 3EG
United Kingdom

Dear Dr. Niakan,

Thank you for submitting your revised manuscript entitled "MICA: A multi-omics method to predict gene regulatory networks in early human embryos". We would be happy to publish your paper in Life Science Alliance pending final revisions necessary to meet our formatting guidelines.

- please upload all figure files as individual ones, including the supplementary figure files; all figure legends should only appear in the main manuscript file
- please add ORCID ID to the secondary corresponding author -- they should have received instructions on how to do so
- please add your main, supplementary figure, and table legends to the main manuscript text after the references section
- please add callouts for Figures 1A; S11A-B; S13A to your main manuscript text

A. FINAL FILES:

B. MANUSCRIPT ORGANIZATION AND FORMATTING:

**Submission of a paper that does not conform to Life Science Alliance guidelines will delay the acceptance of your

manuscript.**

The license to publish form must be signed before your manuscript can be sent to production. A link to the electronic license to publish form will be sent to the corresponding author only. Please take a moment to check your funder requirements.

Sincerely,

October 13, 2023

RE: Life Science Alliance Manuscript #LSA-2023-02415-TRR

Prof. Kathy K Niakan
University of Cambridge
Centre for Trophoblast Research
Department of Physiology, Development and Neuroscience
Cambridge CB2 3EG
United Kingdom

Dear Dr. Niakan,

Thank you for submitting your Resource entitled "MICA: A multi-omics method to predict gene regulatory networks in early human embryos". It is a pleasure to let you know that your manuscript is now accepted for publication in Life Science Alliance. Congratulations on this interesting work.

DISTRIBUTION OF MATERIALS:

Again, congratulations on a very nice paper. I hope you found the review process to be constructive and are pleased with how the manuscript was handled editorially. We look forward to future exciting submissions from your lab.

Sincerely,
